# Quality Evaluation of Bread Prepared from Wheat–Chufa Tuber Composite Flour

**DOI:** 10.3390/foods12030444

**Published:** 2023-01-17

**Authors:** Mehmet Musa Özcan

**Affiliations:** Department of Food Engineering, Faculty of Agriculture, Selcuk University, 42031 Konya, Turkey; mozcan@selcuk.edu.tr; Tel.: +90-332-2232933

**Keywords:** chufa flour, bread, bioactive properties, fatty acids, polyphenols, sensory properties

## Abstract

The oil amounts of breads were measured between 0.13% (control) and 4.90% (with 40% 6 chufa). The total phenolic and flavonoid contents of the breads enriched with chufa tuber flours (powders) were reported as between 37.42 (control) and 99.64 mg GAE/100 g (with 20% chufa) to 61.19 (control) and 120.71 mg/100 g (with 20% chufa), respectively. The antioxidant activities of the bread samples were recorded as between 0.20 (control) and 3.24 mmol/kg (with 20% chufa). The addition of chufa flour caused a decrease in L* values of breads with the addion of tigernut flour. Oleic and linoleic acid contents of the oils extracted from the bread samples enriched with chufa tuber powders were identified as between 61.88 (control) and 66.64% (with 40% chufa) to 14.84% (with 40% chufa) and 17.55% (control), respectively. As a result of the evaluation of sensory properties of breads made from pure wheat flour and composite flours containing 10%, 20%, and 40% chufa tuber flour, the best result was obtained in bread fortified with chufa powder at a concentration of 40%, followed by concentrations of 20 and 10% in decreasing order.

## 1. Introduction

The process of enriching bakery products such as bread, which are rich in carbohydrates and are the most common in our daily food consumption, with functional components is becoming more common day by day [1]. Bread produced by fortification with edible plant derivatives such as fruit and spices is becoming a more functional food in human health [2]. Recently, studies on improving the biological and nutritional values of foods have become widespread by adding various herbs and their products to foods [3,4,5,6]. In order to solve the problem of unbalanced nutrition, which is one of the global problems among people, it is necessary to increase the nutritional and biological value of foods that are widely consumed daily [7]. Recently, success has been achieved in producing more nutritious products using flour obtained from edible roots and legumes, grains, and tubers such as chufa (tigernut) in bakery products [8,9,10]. Bread is an excellent source of nutrients and is available in “ready-made” form. Wheat, the basic ingredient in bread making, is not a widely grown product in most of the countries where bread is consumed, so it may need to be imported, making bread an expensive product. Therefore, the interest in bakery products made with a mixture of agricultural products such as edible seeds, tubers, spices, and roots added to wheat flour is increasing in order to increase the nutritive properties of bread, its health benefits, and consumer interest in dietary fiber [11,12,13,14,15,16]. As chufa has a unique taste in bakery products, it can be used in most bakery products such as delicious cakes and biscuits [17]. Chufa is a tuber rich in carbohydrates, lipids, fiber, some minerals, and vitamins E and C [18,19]. It is thought that chufa flour can be used as a good flour ingredient in the bakery industry. Its consumption is usually fresh, after soaking in water, drying, and roasting. In addition, it is used as a flavoring in ice cream production, and roasted chufa pieces can sometimes be added to biscuits and other bakery products. However, the most common use in the food industry is the production of the product known as “horcata de chufa”. Chufa tubers also contain essential amino acids, starch, and 17–25% oil, which is very close to olive in its properties, depending on the variety [20,21,22,23,24,25,26]. In societies whose eating habits are based on grain products, bread is a basic food source in the daily diet because it is cheap and satisfying. In order to treat and prevent nutritional problems, it is necessary to support the diet and enrich the nutrients. The bread consumed by everyone is enriched by adding different functional components. In order to enrich foods, minerals, vitamins, essential fatty acids, and macro and micro nutrients that must be present in diets must be added. Owing to the loss of important components in wheat during the milling of wheat used in bread production, it is necessary to increase the nutritional value and sensory properties of bread, which is frequently used in the daily diet. Considering the well-documented health benefits of chufa tuber, replacing wheat flour with chufa flour for making bread can improve the nutritional status of the consumer and reduce the use of starch-rich wheat flour. The aim of this study is to investigate the physical, chemical, total phenol, total flavonoid, antioxidant activity, phenolic components, fatty acids, and sensory parameters of breads made from wheat and tiger nut flour mixtures of tiger nut flour substitute at different concentrations instead of wheat flour.

## 2. Material and Methods

### 2.1. Material

The chufa (*Cyperus esculentus*) tubers used in this study were grown in Konya-Sarayönü province. They were sown in May and harvested at the beginning of October 2021. The soil on the surface of the harvested tubers was air-dried after the mud was washed with tap water. The air-dried tigernut tuber samples were brought to the laboratory for analysis.

### 2.2. Methods

#### Preparation of Breads

Here, 100 g flour, 1.5% salt, 1% baker’s yeast, and 60 mL drinking water were used for bread making. After adding chufa powder in concentrations of 10%, 20%, and 40% to wheat flour, the mixture was turned into dough. Dough making was carried out in a laboratory type dough kneading machine. The control group was made without adding chufa flour. After fermentation, the dough of each concentration was kneaded for 5 min and rested for 30 min for intermediate fermantation. After resting, the dough was cut by hand into rounds with a diameter of about 8–10 cm. Each round dough was placed on a greased and fireproof paper tray and baked at 220 °C for 30 min (Figure 1).

### 2.3. Moisture Content

Percent moisture content was determined with the KERN Dbs 60-3 electronic moisture analyzer.

### 2.4. Color Value

The color values of bread samples were made using a Minolta Chroma meter CR 400 (Konica Minolta, Inc. Osaka, Japan). The equipment was calibrated against the white surface calibration plate before measurement and the L*, a*, and b* values, which were determined according to the CIELab color scale [27].

### 2.5. Oil Content

Here, 10 g of dried and ground bread samples were placed in a Soxhlet cartridge and covered tightly with oil-free cotton. Then, after extraction with 250 mL of petroleum ether at 50 °C for 5 h, the balloon containing mycelium was mounted on the evaporator. Petroleum ether was removed at 50 °C in the evaporator and petroleum ether was collected in a separate balloon. The crude oil content remaining in the balloon was calculated gravimetrically (%) [28].

### 2.6. Extraction Procedure

For the extraction procedure, 10 mL of methanol/water (80:20 *v*/*v*) was added to the 1 g ground bread sample. After the mixture was sonicated in an ultrasound bath for 30 min, it was centrifugated at 6000× *g* rpm for 10 min. After the supernatant was leaved, the extract was filtered with a 0.45 µm filter [29].

### 2.7. Total Phenolic Content

Total phenolic contents of bread extracts were determined using the Folin–Ciocalteu reagent according to method described by Yoo et al. [30]. After adding 1 mL of Folin–Ciocalteu and 10 mL of 7.5% Na_2_CO_3_ to the extract, respectively, it was thoroughly mixed with a vortex. The absorbance of extracts was recorded at 750 nm. These results are given as mg gallic acid equivalent/100 g.

### 2.8. Total Flavonoid Content

After 0.3 mL of NaNO_2_, 0.3 mL of AlCl_3_, and 2 mL of NaOH were added to 1 mL bread extract, the mixture was was stirred with a vortex. After that, the absorbance value of solution was recorded at 510 nm. The results obtained were described as mg quercetin/100 g [31]. 

### 2.9. Antioxidant Activity

Free radical scavenging activity of bread extracts was determined using the DPPH according to the method stated by Lee et al. [32]. After the absorbance values of the breads were determined at 517 nm, the results were given as mmol trolox/kg.

### 2.10. Phenolic Compounds

The chromatographic separation of phenolic compounds of bread samples was conducted using HPLC (Shimadzu) equipped with a PDA detector and an Inertsil ODS-3 (5 µm; 4.6 × 250 mm) column. The flow rate of the mobile phase was 1 mL/min at 30 °C. The injection volume was 20 µL. The peaks were obtained at 280 and 330 nm with a PDA detector. The following elution programme was employed: 0–0.10 min 8% B; 0.10–2 min 10% B; 2–27 min 30% B; 27–37 min 56% B; 37–37.10 min 8% B; 37.10–45 min 8% B. The total running time per sample was 45 min. 

### 2.11. Fatty Acid Composition

Fatty acid methyl esters of the bread oils esterificated according to ISO-5509 [33] method with some modifications were analyzed using gas chromatography (Shimadzu GC-2010, Kyoto, Japan) equipped with flame-ionization detector (FID) and capillary column (Tecnocroma TR-CN100, 60 m × 0.25 mm, film thickness: 0.20 µm).

### 2.12. Sensorial Properties

The hedonic test was used to determine the sensory parameters of Ekemke samples. Eight trained panelists were used to determine the sensory characteristics and each panelist evaluated the characteristics of the bread separately by giving the following scores. (1 = very bad, 2 = bad, 3 = fair, 4 = good, 5 = very good).

### 2.13. Statistical Analyses

Data of triplicate analyses were equated and analysis of variance was carried out. The significant variations among the results of chufa tuber powder concentrations were determined by Duncan’s multiple range test (*p* < 0.05).

## 3. Results and Discussion

### 3.1. The Physico-Chemical Properties and Bioactive Properties of Breads Enriched with Chufa 

The physico-chemical properties, total phenol, flavonoid contents, and antioxidant activity values of breads enriched with chufa tuber flours at different levels (10, 20, and 40%) are represented in Table 1. The moisture yields of bread samples decreased from 21.01% (control) to 15.61% (with 20% chufa). Gallagher et al. [34] reported that the moisture content in bread tissue ranged from 3.00% to 5.29%, and they stated that the softness of the bread may be due to the high water content. In another study, the shelf stability of a product to be stored will be better because microorganisms cannot grow at a low moisture content [35]. L* values of breads were found between 56.76 and 76.71, while a* and b* values varied between −0.49 and 5.93 and between 20.45 and 23.70, respectively. The highest L* (76.71) and the lowest a* (−0.49) and b* (20.45) values of breads were measured in the control sample. The addition of tigernut flour caused a decrease in L* values, while an increase was observed in a* and b* values of breads with the addition of tigernut flour (Table 1). Various chemical reactions occurring between protein and carbohydrates are effective on the color values of the bread, and formulation causes differences in the color of the final product [36]. The crumb color (L* values) of bread made with tigernut flour has been reported to be 73.15. In a study reported by Koca and Anil [37], L* and b* values of breads showed a decrease from 58.07 to 26.54 (L*) and from 11.54 to 5.40 (b*) with the addition of hazelnut testa. Similar to the current study, a decrease in L* values from 58.9 to 53.0 and an increase in a* (1.89–5.84) and b* (21.4–27.4) values were observed in breads with the addion of amaranth flour (10, 20, 30, and 40 g/100 g) [38]. L*, a*, and b* values of breads with the addition of chia seed were reported as 67.31–75.7, (−) 0.69–(−) 0.35, and 11.01–13.58, respectively [39]. It is thought to be related to the non-enzymatic caramelization of sugars during cooking [40]. The dark crust color of chufa powder-enriched breads is related to a higher free sugar and amino acid content, and crust color is a result of sugar caramelization and Maillard browning, which is affected by the dispersion of water and the reaction of reducing sugars [40,41]. Moreover, the oil amounts of bread samples were measured between 0.13% (control) and 4.90% (with 40% chufa). Oil contents of cakes mixed with different flour and chufa powder varied between 1.37 and 4.58% [41,42]. This is a result of the high levels of fat found in chufa tuber flour. Because of the protein, which has both hydrophilic and hydrophobic properties, the oil absorption capacity of the bread has increased [43]. The total phenolic substance and flavonoid amounts of the breads incorporated with chufa tuber powders were reported as between 37.42 (control) and 99.64 mg GAE/100 g (with 20% chufa) to 61.19 (control) and 120.71 mg/100 g (with 20% chufa), respectively. In addition, the antioxidant activities of the bread samples were recorded as between 0.20 (control) and 3.24 mmol/kg (with 20% chufa). The results showed some fluctuations depending on chufa tuber powder concentrations compared with the control. The moisture contents of the breads fortificated with chufa flours are decreased in comparison with the control depending on the chufa powder concentration. However, the oil yields of the bread samples increased.

The reduction and increase in the water and oil contents of the bread samples may probably be due to the amount of chufa tuber powder concentrations added. This is because the highest total phenol, flavonoid, and antioxidant activities are detected in the sample of bread enriched with chufa tuber powder at a concentration of 20%. A linear relationship between antioxidant activity and bioactive compounds of bread samples enriched with chufa tuber powders was observed. The highest total phenol, total flavonoid contents, and antioxidant activity values were identified in bread with a 20% concentration of chufa tuber flour, followed by 40% and 10% chufa powders in decreasing order. Statistically significant fluctuations were observed among physico-chemical properties, total phenol, total flavonoid contents, and antioxidant activity values of the wheat breads enriched with chufa tuber powders at different concentrations (*p* < 0.05). While the total phenol content of breads with ginger powder at concentrations of 2%, 4%, and 6% change between 173.1 mgFAE/g and 226.2 mgFAE/g, the antioxidant activity values of ginger breads were found to be between 4.15 and 6.23% [44]. Balestra et al. [45] reported that the antioxidant activity values of breads prepared with ginger flour at concentrations of 2%, 4%, 6%, and 8% increased. An increase in antioxidant activity with chufa tuber powder added to bread was found in the literature, and a similarity was found in the antioxidant property obtained with an increase in chufa powder supplement.

### 3.2. The Phenolic Constituents of the Breads Enriched with Chufa

The phenolic constituents of the wheat breads enriched with chufa tuber powders at different concentrations (10, 20, and 40%) are shown in Table 2. Gallic and 3,4-dihydroxybenzoic acid amounts of the breads fortified with chufa tuber powders at three different concentrations were detected as between 10.95 (with 10% chufa) and 13.98 mg/100 g (control) to 18.16 (control) and 27.05 mg/100 g (with 20% chufa), respectively. Moreover, while catechin amounts of bread samples change between 38.54 (control) and 65.58 mg/100 g (with 20% chufa), rutin values of the breads enriched with chufa tuber flours were identified as between 8.20 (control) and 23.97 mg/100 g (with 40% chufa). In addition, caffeic and syringic acid values of the wheat breads fortified with chufa tuber powders at a certain rate were reported as between 2.01 (control) and 11.06 mg/100g (with 20% chufa) to 2.05 (control) and 5.94 mg/100 g (with 20% chufa), respectively. Kaempferol values of bread samples changed between 1.54 (control) and 7.11 mg/100 g (with 20% chufa). In general, the amounts of phenolic components of the breads increased with the increase in chufa tuber concentrations (Figure 2). Moreover, the highest phenolic component quantities were observed in bread enriched in 20% of chufa powder, followed by 40 and 10% concentrations in decreasing order. Other phenolic constituents were detected at low levels (<4.23 mg/100 g). Statistically significant differences were observed among bioactive compounds of bread samples compared with the control (*p* < 0.05). The reason for the partially low phenolic components of the bread enriched with a concentration of 40% of chufa powder may probably be because of the fact that the high amount of chufa is more damaged by the temperature as a result of the decrease in the flour ratio during baking. Horseradish pomace powder used in biscuit making contained 6.96 (+)-catechin, 0.71 sinapic acid, 1.66 2-hydroxycinnamic acid, and 37.77 mg/100 g rutin [46]. Findings on the bioactive properties of bread samples showed some fluctuations compared with the results of previous studies. The reason for these differences is probably due to the herbal ingredients and their derivatives added to increase the nutritional value and functional properties of the flour.

### 3.3. The Fatty Acid Composition of the Oils of the Wheat Breads Enriched with Chufa Tuber

The fatty acid profiles and their quantitative values of the wheat breads enriched with chufa tuber powders in three different concentrations are presented in Table 3. Palmitic and stearic acid amounts of the oils of the breads enriched with chufa tuber powders changed between 14.56% (with 10% chufa) and 15.00% (with 40% chufa) to 2.35 (with 40% chufa) and 3.76% (control), respectively. In addition, oleic and linoleic acid amounts of the oils of the bread samples enriched with chufa tuber powders were identified between 61.88 (control) and 66.64% (with 40% chufa) to 14.84 (with 40% chufa) and 17.55% (control), respectively (Figure 3). Other fatty acids (myristic, arachidic, linolenic, and behenic acids) detected in bread oils were detected at minor levels (<0.65%). The stearic and linoleic acid contents of the oils extracted from bread enriched with chufa tuber powder have changed together with the increase in chufa concentrations. In addition, an important increase in the oleic acid content of the bread was observed depending on the chufa tuber concentrations. The highest palmitic (15.00%) and oleic acid (66.64%) concentrations were detected in the oil of the bread enriched with chufa power at a concentration of 40%. Statistically significant differences were observed among fatty acid contents of the oils extracted from bread samples with chufa flour compared with the control (*p* < 0.05). Yoon [47] identified 15.2–16.0% palmitic, 2.0–2.2% steraic, 64.4–66.1% oleic, 14.8–15.9% linoleic, 0.4–0.5% linolenic, and 0.5–0.6% arachidic acids in chufa oils extracted by the different solvents (diethyl ether, n-Hexane, and chloroform/methanol). The results were similar compared with the literature values.

### 3.4. The Sensory Properties and Their Points of the Breads 

The sensory properties and their points of the breads evaluated by panelists are illustrated in Table 4. The sensory properties of the breads enriched with different chufa tuber powders showed some differences depending on chufa concentrations. Flavour and smell points of the breads changed between 4.00 (control) and 5.00 (with 20 and 40% chufa) to 4.00 (control) and 4.83 (with 40% chufa), respectively. Moreover, color and texture values of breads were evaluated as between 4.00 (control) and 4.67 (with 10 and 40% chufa) to 3.83 (control) and 4.83 (with 40% chufa), respectively. In general, the most acclaimed bread by the panelists was the bread enriched with 40% chufa powder. The sensory parameters of the breads enriched with chufa tuber powders in different concentrations were highly appreciated compared with the control. Chufa tuber flour improved bread texture and the increase in texture scores for breads with increased chufa flour concentration in wheat flour may have been due to the higher crude fiber content in chufa flour [41]. It has been reported that ginger added to bread at concentrations of 3%, 4.5, and 6% reduces the sensory values of bread [45]. The evaluation of sensory properties of breads made from pure wheat flour and composite flours containing 10%, 20%, and 40% chufa tuber flour shows the potential application of chufa flour in the bakery industry. The most liking was taken to bread fortified with chufa powder at a concentration of 40%, followed by concentrations of 20 and 10% in decreasing order. As a result of the general analysis, the use of wheat flour blended with chufa flour up to 20% gave the best results, and this rate is encouraged in bread making, thus reducing the cost spent on wheat. Sensory evaluation showed that the general acceptability of all bread samples increased in line with the analysis results obtained by increasing the level of chufa tuber flour in bread composite flour.

## 4. Conclusions

The results showed some fluctuations depending on chufa tuber powder concentrations compared with the control. A linear relationship between antioxidant activity and bioactive compounds of bread samples enriched with chufa tuber powders was observed. The highest bioactive properties were identified in bread at the concentration of 20% of the chufa tuber powder added, followed by concentrations of 40% and 10% chufa powders in decreasing order. This study demonstrated the possibility of producing bread of acceptable quality from wheat flour substituted with chufa tuber flour. Wheat flour can be substituted with chufa tuber flour up to an additive level of 20% without any reduction in its sensory properties. Therefore, chufa flours can be recommended to bakers for the production of nutritious and delicious composite breads owing to their high bioactive properties, phenolic components, and fatty acids content. The results of the analysis showed that the nutritional value of the bread increased as the level of chufa flour increased. The sensory outcome indicated that breads made from wheat with 40% chufa flour were more acceptable. It is recommended to use chufa wheat composite flour to reduce the cost of bread production and increase the nutritional value of breads. Bread made from mixture of wheat flour with chufa tuber flour has a good nutritional profile with phytochemicals and nutrients such as oil, bioactive component, antioxidant activity, phenolic compounds, and fatty acids. Chufa flour can be used as a composite flour, defined as a mixture of starch and other ingredients intended to replace wholly or partially wheat flour in bakery and pastry products.

## Figures and Tables

**Figure 1 foods-12-00444-f001:**
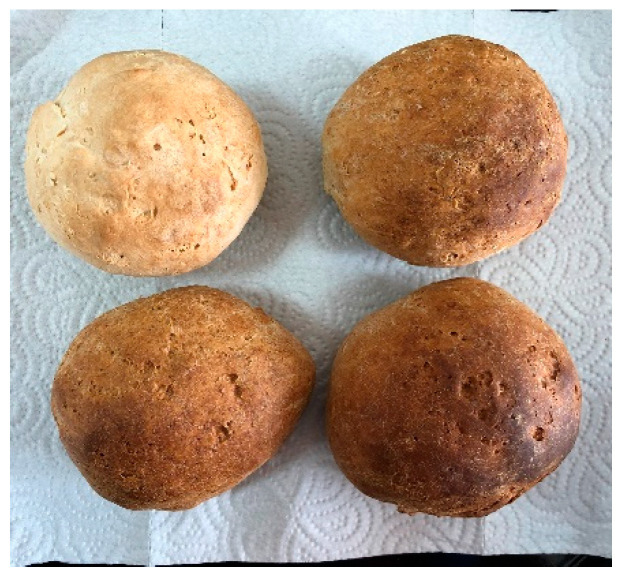
Baked breads enriched with chufa flour.

**Figure 2 foods-12-00444-f002:**
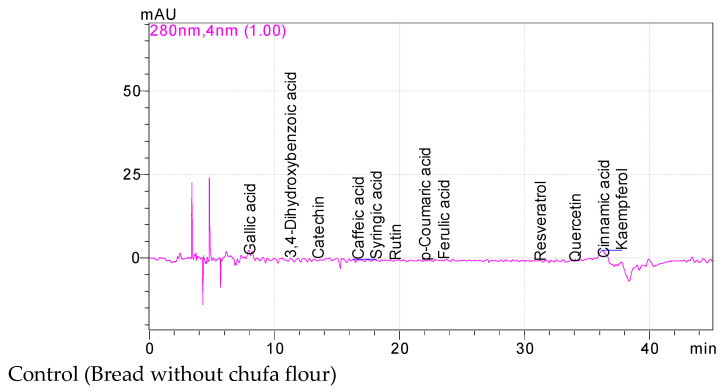
Phenolic chromatograms of the breads enriched with chufa flour at different concentrations.

**Figure 3 foods-12-00444-f003:**
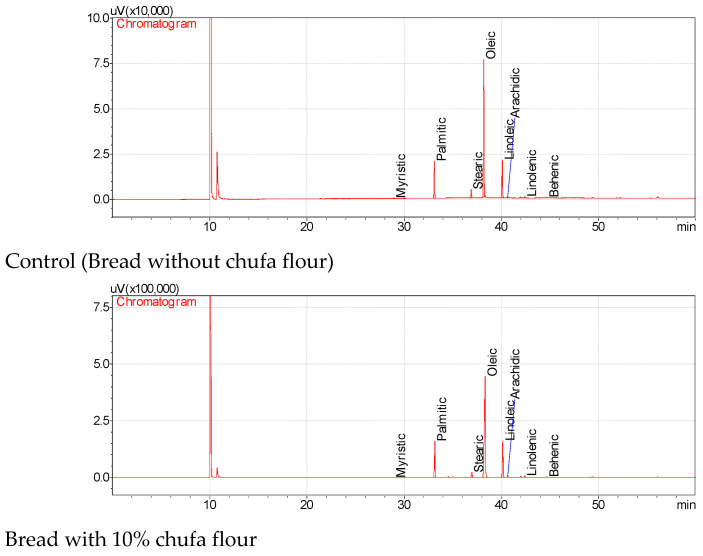
Fatty acid chromatograms of the oils of the breads enriched with chufa flour at different concentrations.

**Table 1 foods-12-00444-t001:** Some chemical and bioactive properties of bread samples enriched with chufa flour at different concentrations *.

Sample	Moisture Content (%)	L	a	b
Control	21.01 ± 2.54 *a	76.71 ± 1.85a	−0.49 ± 0.19d	20.45 ± 0.58d
10%	19.92 ± 0.01c **	66.69 ± 1.86b	3.26 ± 0.03c	23.70 ± 0.03a
20%	15.61 ± 1.57d	56.76 ± 0.28d	5.93 ± 0.37a	22.68 ± 0.26c
40%	20.64 ± 1.57b	58.39 ± 0.66c	5.22 ± 0.43b	23.26 ± 0.35b
Sample	Oil content (%)	Total phenolic content (mg/100 g)	Total flavonoid content (mg/100 g)	Antioxidant activity (mmol/kg)
Control	0.13 ± 0.07d	37.42 ± 6.77d	61.19 ± 0.67d	0.20 ± 0.01d
10%	1.48 ± 0.02c	57.98 ± 3.18c	99.29 ± 3.09c	1.20 ± 0.02c
20%	2.47 ± 0.00b	99.64 ± 6.49a	120.71 ± 3.09a	3.24 ± 0.04a
40%	4.90 ± 0.10a	92.42 ± 1.99b	114.05 ± 4.42b	3.15 ± 0.03b

* Values for control are on a fresh basis, whereas those for other samples are on a dry weight basis. ** Values are mean ± standard deviation and those with different letters in each column are significantly different (*p* < 0.05).

**Table 2 foods-12-00444-t002:** Phenolic compounds of bread samples enriched with chufa flour at different concentrations *.

Phenolic Compounds (mg/100 g)	Control	10%	20%	40%
Gallic acid	13.98 ± 0.48 *a	10.95 ± 0.64d	12.18 ± 0.56b	11.96 ± 0.04c
3,4-Dihydroxybenzoic acid	18.16 ± 0.97d **	19.36 ± 3.00c	27.05 ± 3.60a	20.91 ± 0.57b
Catechin	38.54 ± 2.23d	47.99 ± 0.75c	65.58 ± 4.17a	51.45 ± 0.69b
Caffeic acid	2.01 ± 0.19d	3.55 ± 0.61c	11.06 ± 0.00a	10.21 ± 1.09b
Syringic acid	2.05 ± 0.37d	4.39 ± 0.66c	5.94 ± 0.18a	5.81 ± 0.56b
Rutin	8.20 ± 1.43d	22.86 ± 5.38b	22.67 ± 2.03c	23.97 ± 1.69a
*p*-Coumaric acid	0.40 ± 0.17d	1.12 ± 0.36c	4.23 ± 1.40a	2.31 ± 0.59b
Ferulic acid	0.61 ± 0.13d	2.15 ± 0.88a	1.61 ± 0.30b	0.90 ± 0.33c
Resveratrol	0.48 ± 0.16c	0.73 ± 0.27b	1.15 ± 0.49a	0.22 ± 0.01d
Quercetin	1.60 ± 0.33b	1.22 ± 0.12c	3.79 ± 1.35a	0.55 ± 0.10d
Cinnamic acid	1.00 ± 0.17c	1.00 ± 0.19c	1.88 ± 0.38a	1.74 ± 0.23ab
Kaempferol	1.54 ± 0.11d	2.23 ± 0.51c	7.11 ± 1.78a	2.58 ± 0.24b

* Values for control are on a fresh basis, whereas those for other samples are on a dry weight basis. ** Values are mean ± standard deviation and those with different letters in each row are significantly different (*p* < 0.05).

**Table 3 foods-12-00444-t003:** Fatty acid compositions of the oils extracted from bread samples enriched with chufa flour at different concentrations.

Fatty Acids (%)	Control	10%	20%	40%
Myristic	0.28 ± 0.00a *	0.05 ± 0.00c	ND **	0.06 ± 0.00b
Palmitic	14.97 ± 0.05c	14.54 ± 0.06d	14.90 ± 0.13b	15.00 ± 0.14a
Stearic	3.76 ± 0.03a	2.55 ± 0.00b	2.46 ± 0.03c	2.35 ± 0.01d
Oleic	61.88 ± 0.04d	66.41 ± 0.05b	65.26 ± 0.08c	66.64 ± 0.15a
Linoleic	17.55 ± 0.02a	15.13 ± 0.01c	16.10 ± 0.01b	14.84 ± 0.00d
Arachidic	0.65 ± 0.01a	0.57 ± 0.00b	0.51 ± 0.03c	0.49 ± 0.00d
Linolenic	0.63 ± 0.01a	0.58 ± 0.01b	0.63 ± 0.00a	0.53 ± 0.00c
Behenic	0.29 ± 0.01a	0.17 ± 0.00b	0.13 ± 0.01c	0.13 ± 0.00c

* Values are mean ± standard deviation and those with different letters in each row are significantly different (*p* < 0.05); ** ND: Non detected.

**Table 4 foods-12-00444-t004:** Sensorial properties of bread samples enriched with chufa flour at different concentrations.

Sample	Flavour	Smell	Color	Texture	General View
Control	4.00 ± 0.58c *	4.00 ± 0.58d	4.00 ± 0.58c	3.83 ± 0.69d	3.80 ± 0.40
10%	4.33 ± 0.47b	4.33 ± 0.75c	4.67 ± 0.47a	4.50 ± 0.76c	4.40 ± 0.49
20%	5.00 ± 0.00a	4.67 ± 0.47b	4.50 ± 0.50b	4.67 ± 0.47b	4.80 ± 0.40
40%	5.00 ± 0.00a	4.83 ± 0.37a	4.67 ± 0.47a	4.83 ± 0.37a	5.00 ± 0.00

* Values are mean ± standard deviation and those with different letters in each column are significantly different (*p* < 0.05).

## Data Availability

The data presented in this study are available within the article.

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
