# Peer review of "Quality Evaluation of Bread Prepared from Wheat–Chufa Tuber Composite Flour"

_foods, 2023, doi:10.3390/foods12030444_

Round 1

Reviewer 1 Report

-In title: - Chufa and tigernut are the same names. It would be better if chufa is used in the main title

-In abstract; line 1:” The oil amounts…” should be “The oil contents…”

-Line 2:” The total phenolic substance and flavonoid…” should be “The total phenolic and flavonoid…”

-In Preparation of bread section; 1% leavened? İt should be corrected

-“ml” should be “mL” in text through

-Total phenolic content method should be detailed re-written

-In Phenolic compounds method: Gradient program should be added

-In reference section: DOIs are missing in some references. It should be researched again, and if it is found, it would be better if it was added.

In conclusion, this article fits well with the field of "foods" magazine and I believe it will resonate with readers if published.

Author Response

Response for Reviewer 1:

Comments and Suggestions for Authors

-In title: - Chufa and tigernut are the same names. It would be better if chufa is used in the main title

Answer: Chufa was added instead of tigernut in title.

-In abstract; line 1:” The oil amounts…” should be “The oil contents…”

Answer: :” The oil amounts…” was corrected as  “The oil contents…”

-Line 2:” The total phenolic substance and flavonoid…” should be “The total phenolic and flavonoid…”

Answer: The total phenolic substance and flavonoid…” was revised as  “The total phenolic and flavonoid…”

-In Preparation of bread section; 1% leavened? İt should be corrected

Answer: it was corrected.

-“ml” should be “mL” in text through

Answer: ml was revised as mL in text through.

-Total phenolic content method should be detailed re-written

Answer: Total phenolic content method was detailed re-written

-In Phenolic compounds method: Gradient program should be added

Answer: Gradient program was added

-In reference section: DOIs are missing in some references. It should be researched again, and if it is found, it would be better if it was added.

Answer: DOIs were added to references

In conclusion, this article fits well with the field of "foods" magazine and I believe it will resonate with readers if published.

Reviewer 2 Report

The work deals with the evaluation of breads made with different amount of chufa flour (from 10 to 40%). The novelty of the research is limited to the use of this particular ingredient, but the experimental design and the analyses are well-conducted and useful for the audience. The work is well-written even if some improvement must be done.

·         The abstract should be revised. In this form is simply a list of results, without a minimal introduction.

·         Preparation of breads is very important in the experiment, so more details should be described: i.e. height of the bread, fermentation conditions. Maybe a picture could help.

·         The Author should not conclude that chufa flour can be used for gluten-free products because any tests were carried out with the total replacement of wheat flour.

Some minor points

·         In the abstract the word “chufa” is immediately used, but in the title “tigernut tuber” has been used.

·         Check “tiger nut” or “tigernut”.

·         “mL” instead of “ml” and similar in all the part of the work.

·         L75: remove “was used”.

·         L91: remove “.”.

·         L114: “following scores”, which scores? Add it. Add also scale values information.

·         L116: “the analysis of variance was carried out” instead of “carried out analysis of variance”.

·         L122: add a space between “and” and “40%”.

·         LL124-126: The comparison of gluten-free bread with a very very different moisture is not useful in my opinion.

·         L134: check reference “36”.

·         L140: check “(-)”.

·         LL160-163: sentences are not clear, please reformulate.

·         LL170-175: sentences are not clear, please reformulate.

·         L171: add a space between “173.1” and “mg”.

·         L205: check “(+)”.

·         Tables 2 and 4: check the statistics, i.e. difference between 1.88 ± 0.38 and 1.74 ± 0.23.

·         L234: check “minor”.

·         Figure 2: add a space between “the” and “oils”.

·         L260: maybe “powders” instead of “Powers”.

·         L264: maybe “40%” instead of “20%”.

·         LL261-265: sentences are not clear, please reformulate.

Author Response

Response for Reviewer 2:

Comments and Suggestions for Authors

The work deals with the evaluation of breads made with different amount of chufa flour (from 10 to 40%). The novelty of the research is limited to the use of this particular ingredient, but the experimental design and the analyses are well-conducted and useful for the audience. The work is well-written even if some improvement must be done.

      The abstract should be revised. In this form is simply a list of results, without a minimal introduction.

      Answer: The abstract was revised.

         Preparation of breads is very important in the experiment, so more details should be described: i.e. height of the bread, fermentation conditions. Maybe a picture could help.

        Answer: Bread preparation section has been improved a bit. The picture taken in the experiment has been added.

      The Author should not conclude that chufa flour can be used for gluten-free products because any tests were carried out with the total replacement of wheat flour.

Answer: The conclusion that chufa flour can be used for gluten-free products has been revised, as any test has been completely replaced by wheat flour, and this conclusion is appropriately stated

Some minor points

        In the abstract the word “chufa” is immediately used, but in the title “tigernut tuber” has been used.

        Answer: it was resed as chufa.

         Check “tiger nut” or “tigernut”.

        Answer: it was corrected as tigernut

         “mL” instead of “ml” and similar in all the part of the work.

           Answer: ml was corrected as mL

       L75: remove “was used”.

       Answer:”was used” was removed

         L91: remove “.”.

       Answer: ”.” was removed

         L114: “following scores”, which scores? Add it. Add also scale values information.

          Answer: scores were added.

         L116: “the analysis of variance was carried out” instead of “carried out analysis of variance”.

           Answer: it was revised as “the analysis of variance was carried out.”

         L122: add a space between “and” and “40%”.

Answer: it was corrected

         LL124-126: The comparison of gluten-free bread with a very very different moisture is not useful in my opinion.

      Answer. it was revised partially. Here, the softness of the bread is emphasized due to the high water content.

         L134: check reference “36”.

         Answer:36 th reference was checked.

         L140: check “(-)”.

       Answer: corrected

         LL160-163: sentences are not clear, please reformulate.

          Answer: sentences were reformulated

         LL170-175: sentences are not clear, please reformulate.

        Answer: sentences were reformulated

         L171: add a space between “173.1” and “mg”.

        Answer:a space was added

         L205: check “(+)”.

        Answer: it was corrected

         Tables 2 and 4: check the statistics, i.e. difference between 1.88 ± 0.38 and 1.74 ± 0.23.

        Answer:They were checked and corrected

         L234: check “minor”.

        Answer: it was revised as minor

  • Figure 2: add a space between “the” and “oils”.

          Answer: a space was added

  • L260: maybe “powders” instead of “Powers”.

         Answer: it was corrected as powders

  • L264: maybe “40%” instead of “20%”.

          Answer: it was 40%

  • LL261-265: sentences are not clear, please reformulate.

           Answer: sentences were reformulated

Round 2

Reviewer 2 Report

Use "L" instead of "l" for Liters and similar